

# Improving the prediction of an atmospheric chemistry transport model using gradient boosted regression trees.

Peter D. Ivatt[1,2] and Mathew J. Evans[1,2]

[1]Wolfson Atmospheric Chemistry Laboratories, Department of Chemistry, University of York, York, YO10 5DD, UK
[2]National Centre for Atmospheric Science, Department of Chemistry, University of York, York, YO10 5DD, UK

**Correspondence:** Peter Ivatt (pi517@york.ac.uk)

**Abstract.** Predictions from process-based models of environmental systems are biased, due to uncertainties in their inputs and parameterisations, reducing their utility. We develop a predictor for the bias in tropospheric ozone ($O_3$, a key pollutant) calculated by an atmospheric chemistry transport model (GEOS-Chem), based on outputs from the model and observations of ozone from both the surface (EPA, EMEP and GAW) and the ozone-sonde networks. We train a gradient-boosted decision tree
algorithm (XGBoost) to predict model bias, with model and observational data for 2010-2015, and then test the approach using the years 2016-2017. We show that the bias-corrected model performs significantly better than the uncorrected model. The root mean square error is reduced from from 16.21 ppb to 7.48 ppb, the normalised mean bias is reduced from 0.28 to -0.04, and the Pearson's R is increased from 0.479 to 0.841. Comparisons with observations from the NASA ATom flights (which were not included in the training) also show improvements but to a smaller extent reducing the RMSE from 12.11 ppb to 10.50 ppb, the
NMB from 0.08 to 0.06 and increasing the Pearson's R from 0.761 to 0.792. We attribute the smaller improvements to the lack of routine observational constraints of the remote troposphere. We explore the choice of predictor (bias prediction versus direct prediction) and conclude both may have utility. We show that the method is robust to variations in the volume of training data, with approximately a year of data needed to produce useful performance. Data denial experiments (removing observational sites from the algorithm training) shows that information from one location (for example Europe) can reduce the model bias
over other locations (for example North America) which might provide insights into the processes controlling the model bias. We conclude that combining machine learning approaches with process based models may provide a useful tool for improving performance of air quality forecasts or to provide enhanced assessments of the impact of pollutants on human and ecosystem health, and may have utility in other environmental applications.

# 1 Introduction

Process-based models of the environmental system (e.g. Earth system models and their sub-components) use quantitative understanding of physical, chemical and biological processes to make predictions about the environmental state. These models





typically solve the differential equations that represent the processes controlling the environment, and are used for a range
of tasks including developing new scientific understanding and environmental policies. Given uncertainties in their initial
conditions, input variables, parameterisations etc. these models show various biases which limit their usefulness for some tasks.
Here we focus on predictions of the chemical composition of the atmosphere, specifically on the concentration of tropospheric
ozone ($O_3$). In this region, $O_3$ is a climate gas (Rajendra and Myles, 2014), damages ecosystems (Emberson et al., 2018) and
is thought to lead to a million deaths a year (Malley et al., 2017). The predictions of lower atmosphere $O_3$ from process-based
models are biased (Gaudel et al., 2018), reflecting uncertainties in the emissions of compounds into the atmosphere (Rypdal
and Winiwarter, 2001), the chemistry of these compounds (Newsome and Evans, 2017), meteorology (Schuh et al., 2019) etc.
Understanding and reducing these biases is a critical scientific activity, however, the ability to improve these predictions without
having to improve the model at a process level also has value. For example air quality forecasting, and the quantification of the
impacts air pollutants on human and ecosystem health, would both benefit from improved simulations, even without process
level improvements. Data assimilation techniques (Bauer et al., 2015) are used to incorporate observations into meteorological
forecasts and some air quality forecasts (Bocquet et al., 2015). However, new techniques to improve model predictions would
be useful.

We develop here a method, based on machine learning approaches, to predict the bias (modelled quantity - measured quantity) in a model parameter (in this case tropospheric $O_3$) based on information available from the model and a set of observations
of the parameter. This bias predictor can then be applied more widely (in space or time) to the model output to remove the bias,
bringing the model results closer to reality.

Machine learning has shown utility in the field of atmospheric science, examples include: leveraging computationally burdensome short-term cloud simulations for use in climate models (Rasp et al., 2018), quantifying sea-surface iodine distribution
(Sherwen et al., 2019) and high resolution mapping of precipitation from lower resolution model output (Anderson and Lucas,
2018). More specifically to atmospheric $O_3$, machine learning has been used for improving parameterization in climate models
(Nowack et al., 2018), creating ensemble weighting for forecasts (Mallet et al., 2009) and predicting exposure during forest
fire events (Watson et al., 2019). For bias correction applications, machine learning has been used to correct observational bias
in dust prior to use in data assimilation (Jin et al., 2019).

We describe here the GEOS-Chem model used as our model (Sect. 2), the observations of $O_3$ from four observational
networks (Sect. 3) and our method (Sect. 4) to produce an algorithm to predict the bias in the model. We explore its performance
(Sect. 5) and how it performs under a number of situations and analyse its resilience to a reduction in training data (Sect. 6)
and training locations (Sect. 7). Finally, We explore the choice of predictor in Sect. 8 and discuss the applicability and future
of such a methodology in Sect. 9.

## 2 GEOS-Chem model

For this analysis we use GEOS-Chem Version V11-01 (Bey et al., 2001) an open-access, community, offline chemistry
transport model (http://www.geos-chem.org). In this proof of concept work, we run the model at a coarse resolution of 4°



x 5° for numerical expediency using MERRA2 meteorology from the NASA Global Modelling and Assimilation Office (https://gmao.gsfc.nasa.gov/reanalysis/MERRA-2/). The model has 47 vertical levels extending from the surface to approximately 80 km in altitude. We use the "tropchem" configuration, which has a differential equation representation of the chemistry of the troposphere, and a linearized version in the stratosphere (Eastham et al., 2014). The standard emissions configuration is

used, with EDGAR (Crippa et al., 2018) and RETRO (Hu et al., 2015) inventories for global anthropogenic emissions, which are overwritten by regional inventories where available (NEI (USA) (Travis et al., 2016), CAC (Canada) (van Donkelaar et al., 2008), BRAVO (Mexico) (Kuhns et al., 2005), EMEP (Europe) (van Donkelaar et al., 2008) and MIX (East Asia) (Li et al., 2017)). GFED4 (Giglio et al., 2013) and MEGAN (Guenther et al., 2012) are used for biomass burning and biogenic emissions. Details of the other emissions used and other details of the model can be found online

(http://www.geos-chem.org, (http://wiki.seas.harvard.edu/geos-chem/index.php/HEMCO_data_directories).

To produce the dataset to train the algorithm, the model is run from January 1st 2010 to December 31st 2015 outputting the local model state for each hourly observation (see Sect. 3 and 4). For the testing we run the model from January 1st 2016 to December 31st 2017 outputting the local model state hourly for every grid box within the troposphere.

## 3 Observational dataset

The location of all of the observations used in this study are shown in Figure 1. Ground observations of $O_3$ from the European Monitoring and Evaluation Program (EMEP) (https://www.emep.int), the United States Environmental Protection Agency (EPA) (https://www.epa.gov/outdoor-air-quality-data) and the Global Atmospheric Watch (GAW) (https://public.wmo.int/) are compiled between 2010 and 2018 (see Sofen et al. (2016) for data cleaning). Due to the coarse spatial resolution of this study (4° x 5°), we removed all sites flagged as "urban", as these would not be representative at this model resolution. Similarly,

all mountain sites (observations made at a pressure <850 hPa ) were removed due the difficulty in representing the complex topography typical of mountain locations within the large grid boxes. Ozone-sonde data from the World Ozone and Ultraviolet Radiation Data Centre was also used (https://woudc.org). Ozone-sonde observations above 100 ppb of $O_3$ were excluded as they are considered to be in the stratosphere (Pan et al., 2004). For both surface and sonde observations, when multiple observations were found in the same hourly model grid-box (both in the horizontal and vertical) they were averaged (mean) together to

create a single "meta-site". There are 13,118,334 surface meta-site observations in the training period between 1/1/2010 to 31/12/2015, and 3,783,303 in the testing period between 1/1/2016 and 31/12/2017. There are 250,533 ozone-sonde meta-site observations in the training period and 78,451 in the testing.

Observations of $O_3$ from the NASA Atmospheric Tomography Mission (ATom) flights (Wofsy et al., 2018) were used as an independent testing data set. ATom flew over the Pacific and the Atlantic from the northern mid-latitudes to the southern

and back from the surface to 15 km measuring the concentration of many compounds including $O_3$ (Figure 1). It flew for each of the four seasons between July 2016 and May 2018, but only the first three (summer, spring and winter) are used due to availability at the time of writing. Given the oceanic nature of the flights and their sampling through the lowermost 15 km of the atmosphere, the observations collected are most typical of sonde observations. As with the sonde data, any $O_3$





observations greater than 100 ppb was removed, and data were averaged onto the model grid resolution (mean) to give hourly

model resolution "meta" sites. Once averaged, there are 10,518 meta observations used for the algorithm testing.

## 4   Developing the bias predictor

To develop a predictor for the bias in the model $O_3$ we use the hourly observations from the surface and sondes for the training

period (1/1/2010 to 31/12/2015). We run the model for the same period, outputting values of the model's local "state" at

each observation location in space and time. The model local state consists of the grid box concentration of the 68 chemicals

transported by the model (including $O_3$) and 15 physical model parameters (see Table 1).

Once each $O_3$ observation has a corresponding model prediction, we can develop a function to predict the model bias given

the values of the model local state as input. Several potential "machine learning" methodologies exist for making this prediction,

including neural nets (Gardner and Dorling, 1998) and decision trees (Breiman, 2001). We have tended to favour decision tree

methods due to their increased level of explicability over neural nets.

As with other machine learning approaches, decision tree techniques (Blockeel and De Raedt, 1998) make a prediction for

the value of a function based on a number of input variables (features) given previous values of the function and the associated

values of the features. It is essentially non-linear multi-variant regression. A single decision tree, is a series of decision-nodes

that ask whether the value of a particular feature is higher than a specific value. If the value is higher, progress is made to

another decision-node, if it is not, progress is made to a different decision-node. Ultimately, this series of decisions reaches a

leaf-node which gives the prediction of the function. The depth of a tree (the number of decision needed to get to a leaf-node)

is an important aspect of tuning decision trees. If the tree is too shallow it will miss key relationships in the data. Conversely,

if a model is too deep it will over-fit to the specific dataset and will not generalise well. The training of the system relies upon

deciding which features should be used by each decision node and the specific value to be tested. The use of a single decision

tree leads to over-fitting (Geurts et al., 2009), so this progressed to using random forest regression (Breiman, 2001), where a

number of decision trees are constructed with differing sampling of the input data. The mean prediction of all of the decision

trees (the forest) was then used as the prediction of the function. More recently, gradient boosting regression (Friedman, 2002)

relies on building a tree with a relatively shallow depth and then fitting a subsequent tree to the residuals. This this is then

repeated until an adequate level of complexity is reached.

The gradient boosted regression technique suited our needs for a variety of reasons: it is able to capture non-linear relation-

ships which underlies atmospheric chemistry (Gardner and Dorling, 2000); the decision tree-based machine learning technique

is more interpretable than neural net-based models (Kingsford and Salzberg, 2008); it has a relatively quick training time al-

lowing efficient cross validation for tuning of hyper parameters; and it is highly scalable meaning we are able to test on small

subsets of the data before increasing to much longer training runs (Torlay et al., 2017). For the work described here we use the

XGBoost (Chen and Guestrin, 2016; Frery et al., 2017) algorithm.

Hyper-parameters are parameters set before training that represent the required complexity of the system being learnt

(Bergstra and Bengio, 2012). Tuning of these parameters was achieved by five kfold cross validation whereby the training





data is broken into five subsets, with the training data organised by date. The model was then trained on four of these sub-sets and tested on the remaining subset. Training and test is repeated on each of the five subsets to identify the optimum hyper-parameters attempting to balance complexity without over fitting (Cawley and Talbot, 2010).

The key hyper-parameters tuned were the number of the trees and depth of trees. Similar results could be found with 12 to 18 layers of tree depth, with a reduction in number of trees needed at greater depth. It was found that the algorithm achieved the majority of its predictive power early on, with the bulk of the trees producing small gains in root mean square error. As a compromise between training time and predictive strength, 150 trees with a depth of 12, were chosen. This took 1 hour to train on a 40 core CPU node, consisting of two Intel Xeon Gold 6138 CPUs. Mean squared error was the loss function used

for training.

Numerous model performance metrics are used in subsequent assessment of the model performance. The Root Mean Squared Error (RMSE) measures the average error in the prediction, Normalised Mean Bias (NMB) measures the the direction of the bias and normalises the mean value, the Pearson's R correlation coefficient measures the linear relationship between the prediction and the observation.

$$RMSE(y, \hat{y}) = [\frac{1}{N} \sum_{i=1}^{N} (y_i - \hat{y}_i)^2]^{\frac{1}{2}} \tag{1}$$

$$NMB(y, \hat{y}) = \frac{\sum_{i=1}^{N} (y_i - \hat{y}_i)}{\sum_{i=0}^{N} y_i} \tag{2}$$

$$R(y, \hat{y}) = \frac{\sum_{i=1}^{N} [(y_i - \bar{y}_i)(\hat{y}_i - \bar{\hat{y}}_i)]}{\sum_{i=1}^{N} [(y_i - \bar{y}_i)^2 (\hat{y}_i - \bar{\hat{y}}_i)^2]^{\frac{1}{2}}} \tag{3}$$

Where $y$ is the observed values, $\hat{y}$ is the predicted values and $N$ is the number of samples.

## 5   Application

With the bias predictor now trained we can now apply it to the model output and evaluate performance. We do this for a different period (1/1/2016-31/12/2017) to that used in the training (1/1/2010-31/21/2015). We first look at the mean daily (diurnal) cycles calculated the model for nine globally distributed sites (Figure 2 with statistics given in Table 2). The base

model (blue) shows significant differences with the observations (black) for most sites. The subtraction of the bias prediction from the base model (red), leads to a significant increase in the fidelity of the simulation. For the US sites, the base model over estimates at all times, consistent with previous work (Travis et al., 2016), with the largest biases occurring during the night. The bias corrected model now shows a diurnal cycle very similar to that observed, with Rs increasing from a mean of 0.916 to 0.997, RMSEs reducing from a mean of 15.1 ppb to 1.09 ppbv, and NMB reducing from a mean of 0.51 to -0.02. The bias correction

thus successfully corrects biases seen in the mean diurnal cycle, notably the large night-time bias. Although the base model failure is less evident for the European sites (Hu et al., 2018), there are still in general small improvements with the inclusion





of the bias corrector. The Japanese data shows a differing pattern. Similar to the US sites, the base model over-estimates the $O_3$ generating a much smaller diurnal cycle compared to the observation. Although the bias corrector improves the mean value, it does not completely correct the diurnal cycle. We attribute this to the coastal nature of Japan. The model grid-box containing
the Japanese observations is mainly oceanic but the observations show a continental diurnal cycle (a significant increase in $O_3$ during the day similar to those seen in the US). If there is a fundamental mismatch between the model's description of the site and the reality (ocean vs land), the bias-predictor is unable to completely remove the bias. For the two clean tropical sites (Cape Verde and Cape Point in South Africa) the base model already does a reasonable job (Sherwen et al., 2016) so the bias corrected version improves little and slightly reduces the NMB performance at Cape Verde from 0.03 to 0.04. For the Antarctic
site the large bias evident in the model (Sherwen et al., 2016) is almost completely removed by the bias corrector but that results in a small reduction in the R value.

The seasonal comparison (Figure 3 with statistics given in Table 3) shows a similar pattern. Over the polluted sites (USA, UK, Germany) biases are effectively removed. The performance for Japan is less good, with the clean tropical sites again showing only small improvements. Over Antarctica a significant bias is removed with the application of the bias corrector. Where the
performance of the model is already good, such as the RMSE at Cape Verde, or for the NMB at the UK the inclusion of the bias correction can slightly degrade performance.

A point by point comparison between all of the surface data (1/1/2016-31/12/2017) and the model with and without the bias corrector is shown in Figure 4. The bias corrector removes virtually all of the model biases (NMB) taking it from 0.29 to -0.04, significantly reduces the error (RMSE) from 16.21 ppb to 7.48 ppb and increases the correlation (Pearson's R) from 0.479 to
0.841. Although this evaluation is for a different time period than the training dataset, it is still for the same sites. It would be preferable to use a completely different dataset to evaluate the performance of the system.

We use the ATom dataset to provide this independent evaluation. Figure 5 (with statistical data in Table 4) shows the comparison between the model prediction of the ATom observations with and without the bias corrector. Although the inclusion of the bias correction improves the performance of the model, this improvement is significantly smaller than that seen for the
surface data. The RMSE is reduced by only 13% for the ATom data compared to 54% for the surface observations. Similarly the Pearson's R only marginally improves with the use of the bias corrector. Much of the improvement of the model's performance for the ATom data will be coming from the observations collected by the sonde network. There are significantly fewer observations (40:1) collected by that network than by the surface network. Thus for the bias corrector approach to work well it appears that there must be significant volumes of observations to constrain the bias under sufficiently diverse conditions. It
would appear that the sonde network may not provide that level of information to the degree that the surface network does.

Applying the bias corrector to all of the grid points within the model shows the global magnitude of the predicted bias (Figure 6). Similar to the analysis of the nine individual sites, the base model is predicted to be biased high over much of the continental USA, with smaller biases over Europe and the tropical ocean regions. Over the southern ocean the model is predicted to be biased low. However, the bias is also predicted for regions without observations (see Figure 1). For example,
over China, the model is predicted to be biased high by ~15 ppbv. This is higher but not dissimilar to the biases previously found for the model in China (Hu et al., 2018) which found a positive bias of 4-9 ppbv but using a different model configuration





(higher resolution) and for a different model assessment (MDA8 vs annual mean). Similar questions as to the accuracy of the prediction arise from the large biases predicted for central Africa and South America. Future evaluation of the bias corrector methodology should more closely look at the impact on these regions and where possible extend the training dataset to use observations from these regions if they are available.

The gain (the loss reduction gained from splits using that feature (Chen and Guestrin, 2016)) is shown in Figures 7. This provides a diagnostic of the importance of different input variables in the decision trees used for making predictions. Surprisingly, the most important feature from this analysis is the concentration of $NO_3$ (the nitrate radical). This has a high concentration in polluted night-time environments and low concentration in clean regions or during daytime (Winer et al., 1984). This feature appears to be being used to correct the concentration of $O_3$ in regions such as the US which are polluted and have a significant high bias at night. The next most important feature is the $O_3$ concentration itself. This may reflect biases in regions with very low $O_3$ concentrations such as around Antarctica. The third most important feature is the $CH_2O$ concentration. This may indicate biases over regions of high photo-chemical activity, as $CH_2O$ is a product of the photo-chemical oxidation of hydrocarbons (Wittrock et al., 2006). Future work should explore these explanatory capabilities to understand why the bias correction is performing as it is. This may also allow for a scientific understanding of why the model is biased rather than just how much the model is biased.

We have shown that the bias corrector method provides an enhancement of the base-model prediction under the situations explored. We now perform some experiments with the system to explore its robustness to the size of the dataset used for training both spatially and temporally.

## 6   Size of training dataset

The bias predictor was trained using six years of data (2010-2015). This provides a challenge for incorporating other observational data sets. For some critical locations such as China or India the observational record is not that long and for high resolution model data (eg 12.5 km (Hu et al., 2018)) managing and processing 73 parameters for six years could be computationally burdensome. Being able to reduce the number of years of data whilst maintaining the utility of the approach would therefore be useful. Figure 8 shows the improvement in the global performance of the model metrics (same as for Table 4) for surface $O_3$ varying the number of months of training data used. The end of the training set was the 1[st] of January 2016 in all cases and the starting time pushed backwards to provide a sufficiently long training dataset. The dot in Figure 8 represent the statistical performance of the uncorrected model. Training with only a month of data (in this case Dec 2015) marginally reduces the RMSE and the Pearson's R. However, it causes a change in the sign of the NMB, as the model's winter time bias is projected over the whole year. Significant benefit arises once at least eight months of training data has been included. Using a bias predictor trained with a year of observational data increases the performance of the base model, halving the RMSE, removing most of the NMB and increasing the Pearson's R by 60%. Much of the variability in the power-spectrum of surface $O_3$ is captured by timescales of a year or less (Bowdalo et al., 2016) thus a timescale of a year appears to be a good balance between computational burden and utility for an operational system such as air quality forecasting.



## 7 Data denial

Now we explore the impact of removing locations from the training dataset. We start by removing the data from the nine meta sites shown in Figures 9 and 10 (California, New York, Texas, UK, Germany, Japan, Cape Verde, South Africa (Cape Point), Antarctica (Neumayer)) from the algorithm training dataset (again for 2010-2015) and evaluate the bias corrected model using this new bias predictor for these sites (again for 2016-2017). In the USA, removing the nine observational data sets does degrade the overall model performance slightly (the green lines in Figures 9 and 10) compared to the full training dataset (red line). It appears that the neighbouring sites are similar enough to the removed sites to provide sufficient information to almost completely correct the bias. There are different degrees of impact for the other sites. For the UK, the impact of removing the UK site from the training dataset is minimal. For Germany, the bias corrections are now larger, and over compensates the base model during the night and in the summer months. For Japan the removal of its information provides a simulation halfway between the simulation with and without the standard bias correction. For remote sites, such as Cape Verde and South Africa, removal makes the bias corrected model worse than the base model. Similar to Japan, removing the Antarctic site leads to a bias correction which is between the standard bias corrected model and the standard model. A full set of statistics for the diurnal and seasonal results can be found in Tables 2 and 3 respectively.

Much of this behaviour relates to the similarity of other sites in the training dataset to those which were removed. For sites such as the US, and to some extent Europe, removing a few sites has little influence on the bias predictor as there are a number of similar neighbouring sites which can provide that information. For other locations such as the clean Cape Verde and South African sites there are no other similar sites. Thus removing those sites from the training dataset removes significant amounts of information. If there are no similar sites for the bias correction to use, an inappropriate correction can be applied which makes the simulation worse. For sites such as the Japanese and Antarctic sites there are some similar sites in the training data to provide some improvement over the base model.

Taking the data denial experiments further, we remove all observations within North and South America from the training dataset (everything between -180° and -10° East). Figures (11 and 12) show the impact of this on the standard nine sites. For New York and Texas the bias corrected model performs almost as well without North and South America as it does with. The bias corrector predicts roughly the same correction for California as it does for New York and Texas and this over-corrects daytime concentrations for California but simulates the night time and the seasonal cycle much better than without the bias corrector. For the other six sites around the world, the influence of removing North and South America is minimal. It appears surprising that the corrections applied for North America are so good even though the North American data is not included within the training. This suggests that at least some of the reasons for the biases in the model are common between, say North America and Europe, indicating a common global source of some of the bias. This may be due to errors in the model's chemistry or meteorology, which would be global rather than a local source of bias.

Figure 13 shows the changes in prediction that would occur globally if the western hemisphere($-180^o$E to $-10^o$E) is removed from the training data. Where there are observations in the eastern hemisphere, changes are in general small. But there are some significant changes for locations that do have observations such as in Spain. It appears the algorithm is using information from



the North American observations to infer corrections for Spain. These are relatively similar locations (photolysis environment,
temperatures, emissions etc) and so the algorithm is using information from North America in the Spanish predictions. The
difference in these predictions may suggest that there are different causes in the biases between the North American sites and
the Spanish sites. The changes are much more profound in areas that have no observations of their own to constrain the problem.
Removing the Western hemisphere reduces the number of unique environments the algorithm has to learn from resulting in
significant changes in the prediction.

These types of data denial experiments may in the future provide an ability to explain model failings which could be used to
help improve the process level representation within models.

## 8    Nature of the prediction

The bias correction method described here, attempts to predict the bias in the model. An alternative approach would be to
directly predict the $O_3$ concentration. An algorithm to do this given the same model local state information is trained on the
standard six years of training data (2010 - 2015). Table 4 shows a statistical analysis of the performance for the model, coupled
to both the bias predictor and the direct predictor. For the testing years (2016 to 2017) the direct prediction of surface $O_3$
performs marginally better than the bias correction for some metrics (RMSE of 7.11 ppb versus 7.48 ppbv, NMB of 0.00 vs
-0.04, and R of 0.850 versus 0.841) but for some metrics the performance is less good (Slope of best fit of 0.84 versus 0.89 and
a y-intercept of 4.96 ppbv versus 2.07 ppbv). However, for the ATom dataset, the bias predictor performs better (Table 4). We
interpret this to mean that for locations where observations are included in the training (surface sites and sondes), directly using
those observed has benefits. However for sites where no observations are used, it is better to use the bias corrector approach.
Further work is necessary to advance our understanding of the form of the prediction that is necessary to best provide a useful
enhancement of the system.

## 9    Discussion

We have shown that the bias in the $O_3$ concentration calculated by a chemistry transport model can be reduced through the
use of a machine learning algorithm with the results appearing robust to data denial and training length experiments. For
activities such as air quality forecasting for sites with a long observational record this appears to offer a route to significant
improvements in the fidelity of the forecasts without having to improve process level understanding. This work offers some
practical advantages over data assimilation. The observations don't necessarily need to be available in real time as the training
of the bias predictor can be made using past observations and applied to a forecast without the latest observations being
available. The approach may also be applied to regions where observational data is not available. Although this necessitates
care, the temporary lack of availability of data is much less of a problem for this approach than for data assimilation.

Significantly more future work is needed to understand the approach than has been shown in this proof of concept work.
Exploring the number and nature of the variables used would thus be advantageous. The complete set of model tracers and some





physical variables were used here but their choice was somewhat arbitrary. A more systematic exploration of which variables are needed to be included is necessary. Are all the variables needed? Are important physical variables missing? Similarly, only one machine learning algorithm has been used with one set of hyper-parameters chosen. Algorithm development is occurring very quickly, and we have not explored other approaches such as neural nets that may offer improved performance. The ability to predict the bias for regions without observations is also a potentially useful tool for better constraining the global system.

Observations of surface $O_3$ exist for China (Li et al., 2019) but have not been included here for expediency. It would be scientifically interesting to see how they compare to those predicted by the bias corrector and how the bias corrector changes if they are included in the training. It seems possible that the approach developed here could be used to explore methods to extract information about why the model is biased rather than just quantifying that bias. Some hint of that is given by the importance of the nitrate radical ($NO_3$) in the decision trees which highlights the night time as being a large factor in the model

bias. Finally, the method could readily be extended to other model products such as PM2.5.

More generally machine learning algorithms appear to offer significant opportunities to understand the large, multivariate and non-linear data sets typical of atmospheric science and the wider environmental sciences. They offer new tools to understand these scientifically interesting, computationally demanding and socially relevant problems. However, they must also be well characterised and evaluated before they are routinely used to make the forecasts and predictions.

*Code and data availability.*

The GEOS-Chem model code is available from https://github.com/geoschem/geos-chem and the XGBoost code used is available from https://xgboost.readthedocs.io. Licensing agreements mean that we are unable to redistribute the observational data however it is all publicly available.

The GAW $O_3$ data is available from http://www.wmo.int/pages/prog/arep/gaw/world_data_ctres.html.

The EMEP $O_3$ data is available from http://ebas.nilu.no.

The EPA $O_3$ data is available from https://www.epa.gov/outdoor-air-quality-data.

The ozone-sonde data is available from https://doi.org/doi:10.14287/10000008.

The ATom data is available from https://doi.org/10.3334/ornldaac/1581.

*Author contributions.*

Both authors contributed equally to the development and writing of this paper.

*Competing interests.*

There are no competing interests.





*Acknowledgements.* This project was undertaken on the Viking Cluster, which is a high performance compute facility provided by the University of York. We are grateful for computational support from the University of York High Performance Computing service, Viking and the Research Computing team. We also acknowledge funding from the Natural Environment Research Council (NERC) through the "Big data for atmospheric chemistry and composition: Understanding the Science (BACCHUS)" (NE/L01291X/1) grant.

We thank: the numerous individuals and organisations responsible for delivering the GAW, EPA and EMEP observations for their efforts and dedication; Tom Ryerson, Jeff Peischl, Chelsea Thompson, and Ilann Bourgeois of the NOAA Earth System Resources Laboratory's Chemical Sciences Division for their effort in collecting the ATom observations.

We thank the the National Centre of Atmospheric Science for funding for Peter Ivatt through one of its Air Quality and Human Health studentships.



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

470





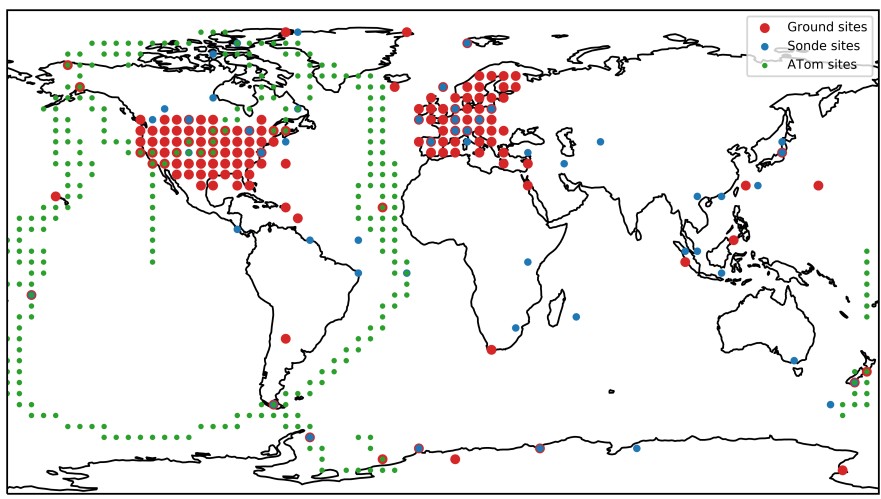

**Figure 1.** Locations of meta observations (averaged over model 4°x5° grid boxs) from the surface (EPA,EMEP and GAW indicated in red), the ozone-sonde network (blue) an the ATom flights (Green).



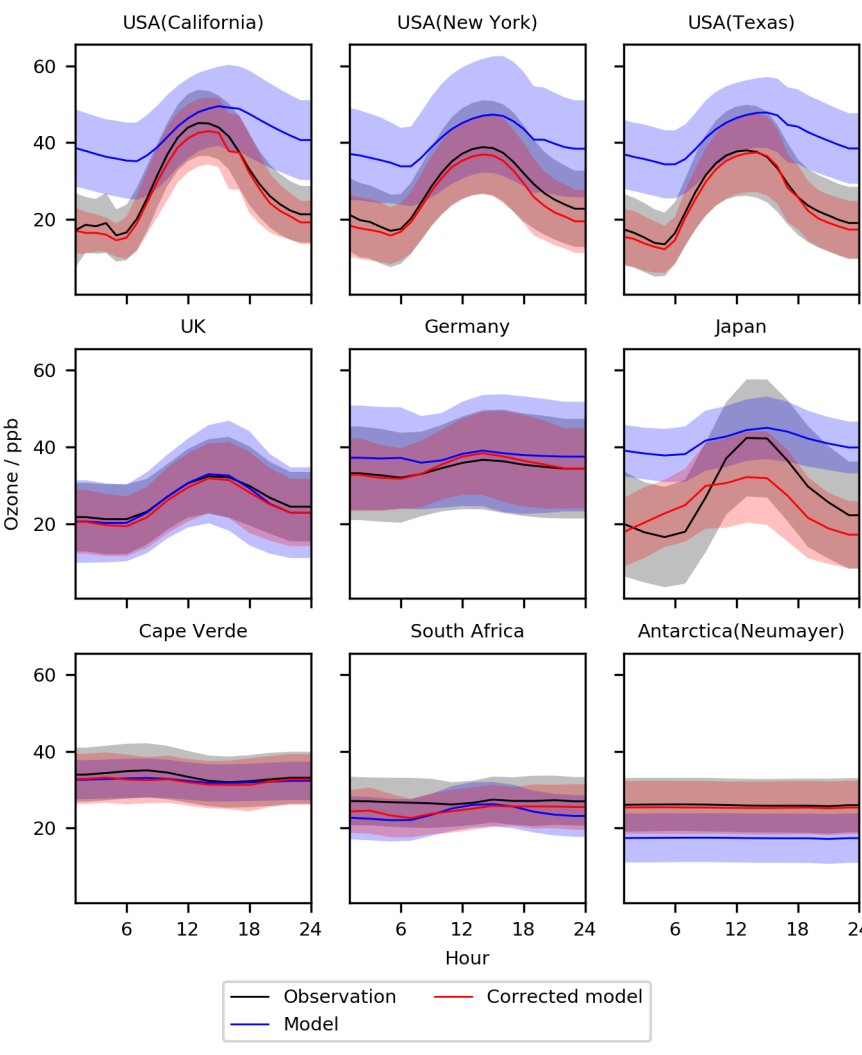

**Figure 2.** Diurnal cycle for $O_3$ at nine meta sites in 2016-2017. Shown are the observations, the base model and the model corrected with the bias predictor. The median values are shown as the continuous line and the $25^{th}$ to $75^{th}$ percentiles as shaded areas.

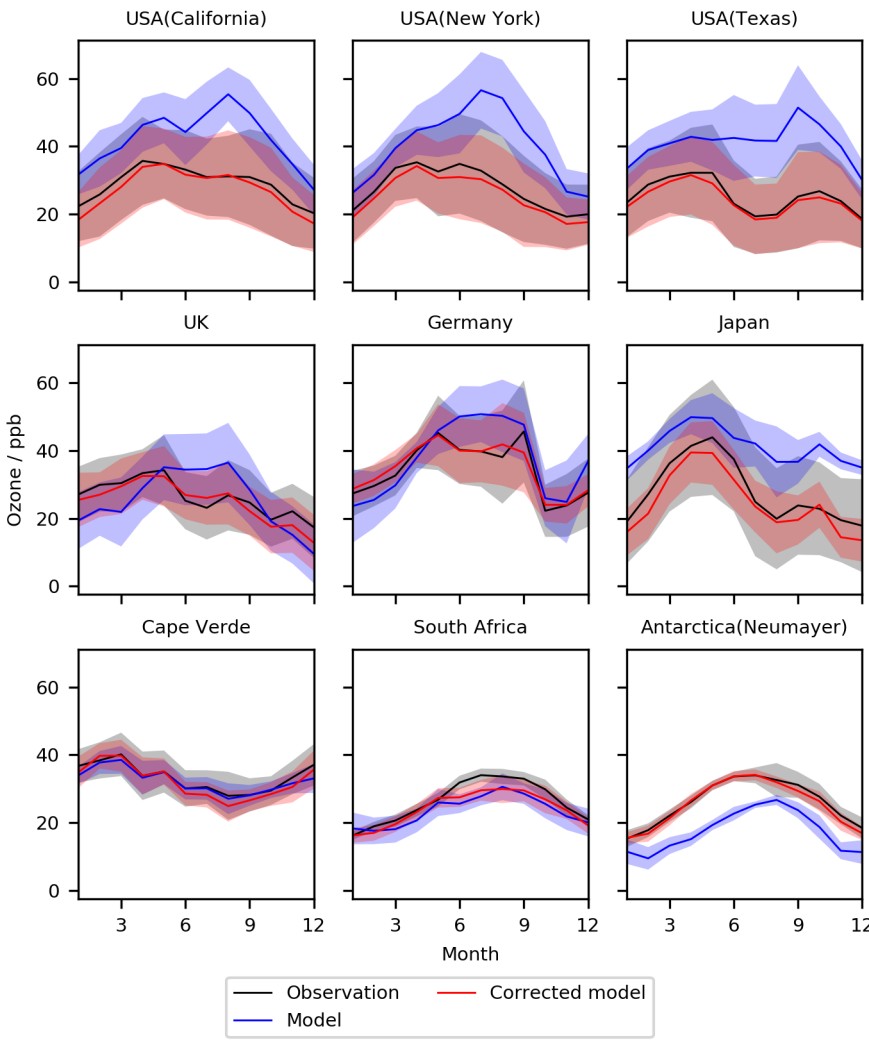

**Figure 3.** Seasonal cycle for O$_3$ at nine meta sites in 2016-2017. Shown are the observations, the base model and the model corrected with the bias predictor. The median values are shown as the continuous line and the 25$^{th}$ to 75$^{th}$ percentiles as shaded areas.

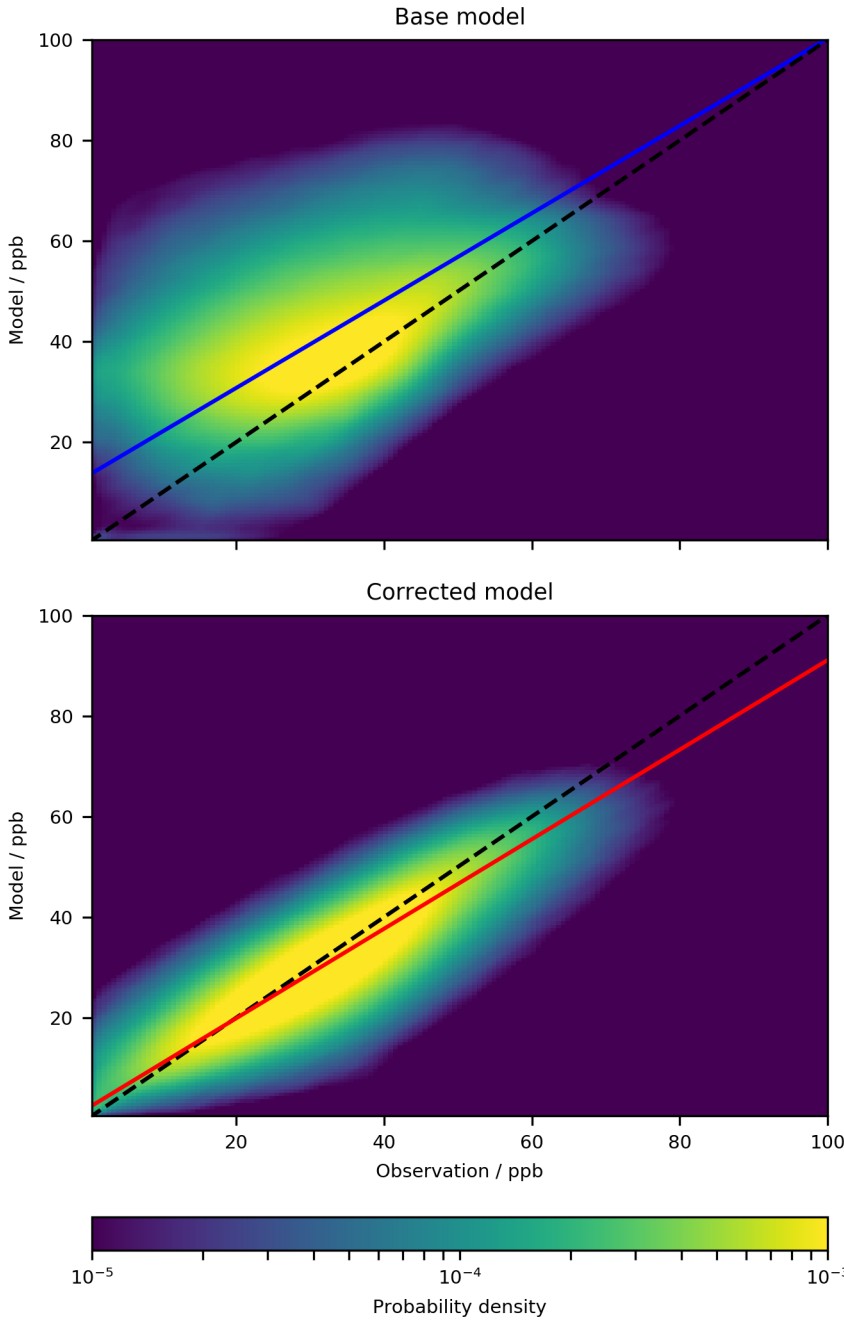

**Figure 4.** Kernel density estimation plot of model vs observation for all ground sites in the model (upper panel) and corrected model (lower panel) for 1/1/2016 to 31/12/2017. The plot is made up of 3,783,303 data points.

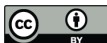

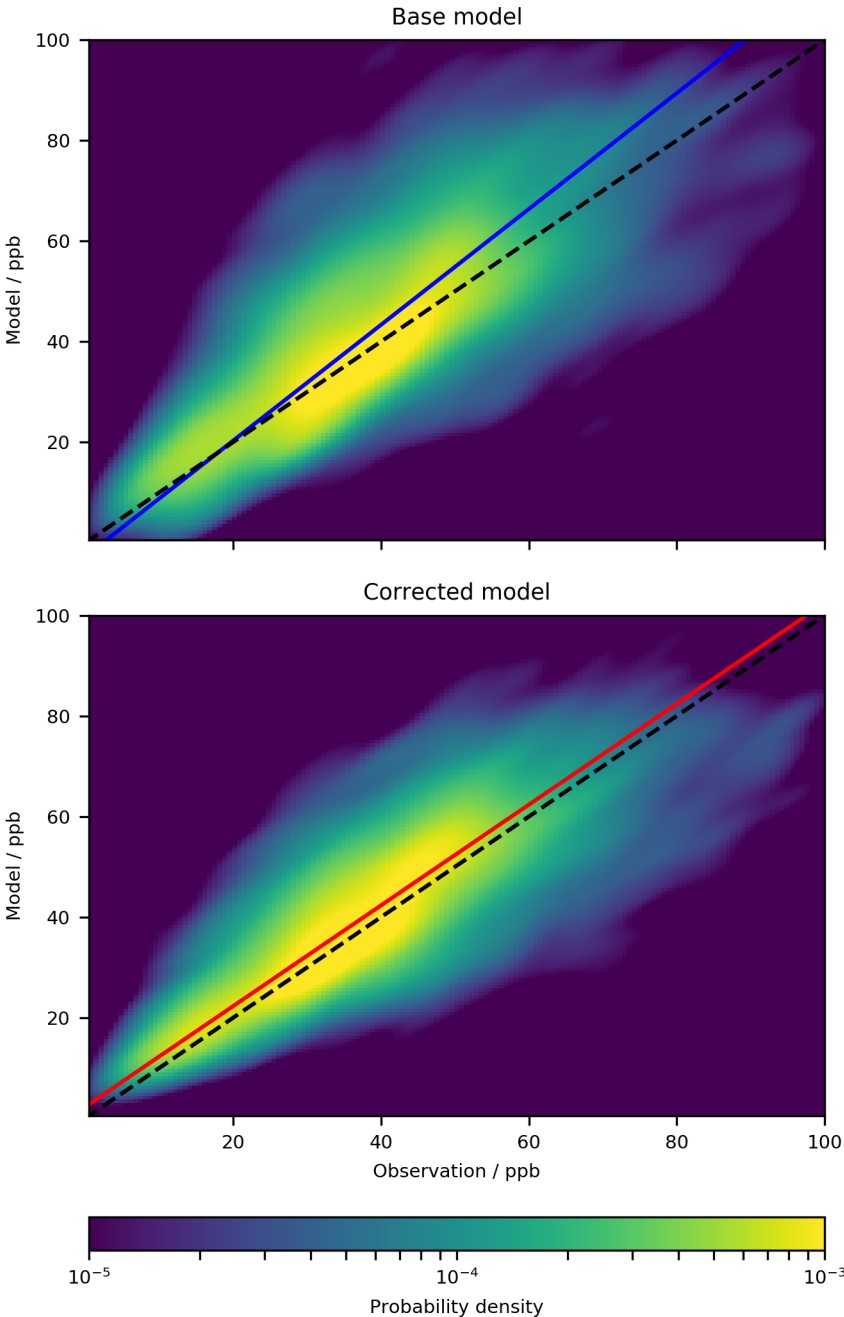

**Figure 5.** Kernel density estimation plot of model vs observations for all ATom summer, winter and fall campaign observations compared to model (upper) and corrected model (lower). The plot is made up of 10,518 data points.





**Figure 6.** Annual mean predicted bias (model - measurement) that would be applied to all grid boxes for a one year (2016) model simulation in three areas of the atmosphere. The >100 ppb of $O_3$ definition of the stratosphere is used.

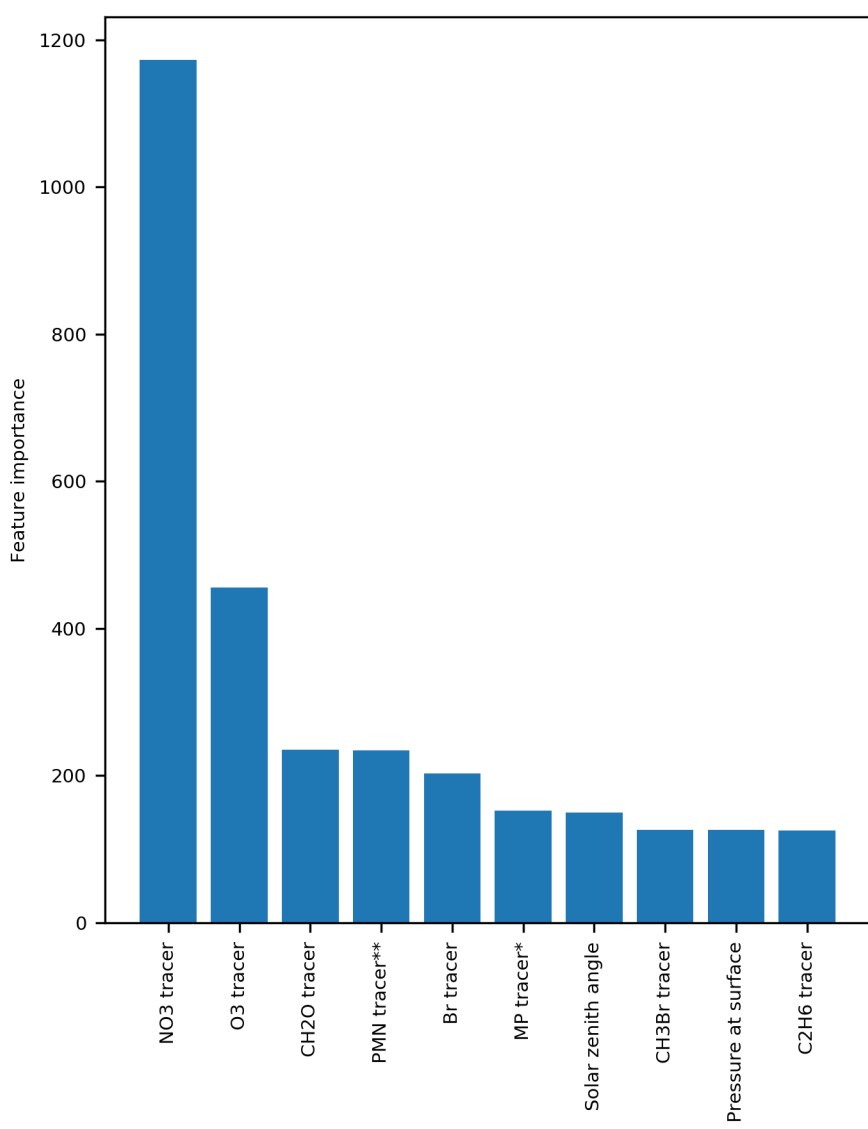

**Figure 7.** Feature importance based on gain (the average gain across all splits the feature is used in). *Methyl-hydro-peroxide, **Peroxymethacryloyl nitrate

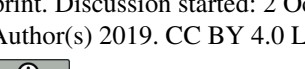



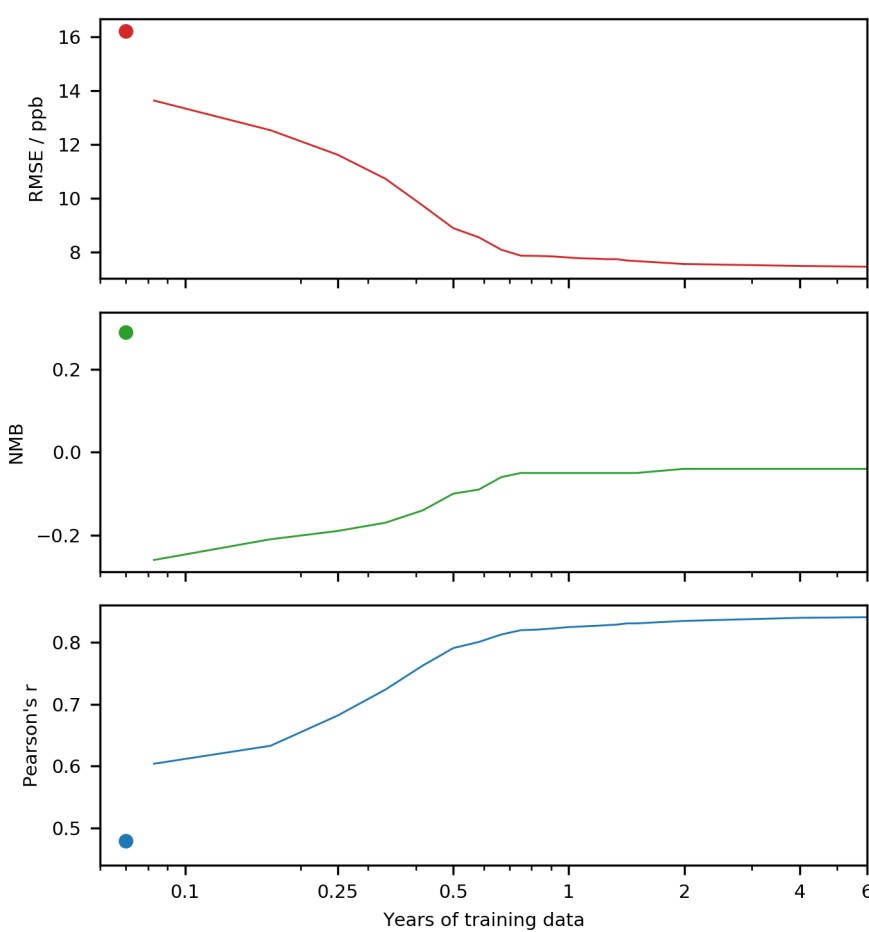

**Figure 8.** Testing statistics with increasing length of training data. The dot represents the uncorrected model performance.

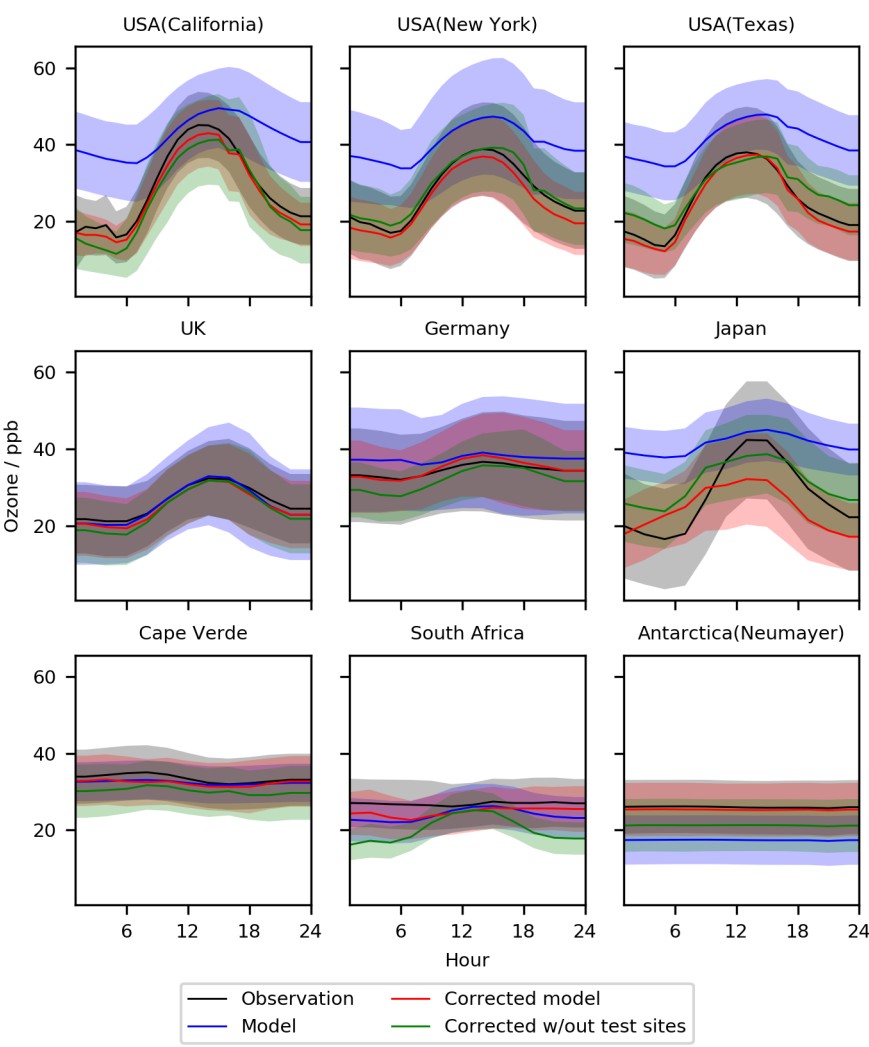

**Figure 9.** Diurnal cycle for O₃ at nine meta sites in 2016-2017. Shown are the observations, the base model and corrected model trained with all observations occurring at the nine sites removed. The median values are shown as the continuous line and the $25^{th}$ to $75^{th}$ percentiles as shaded areas.

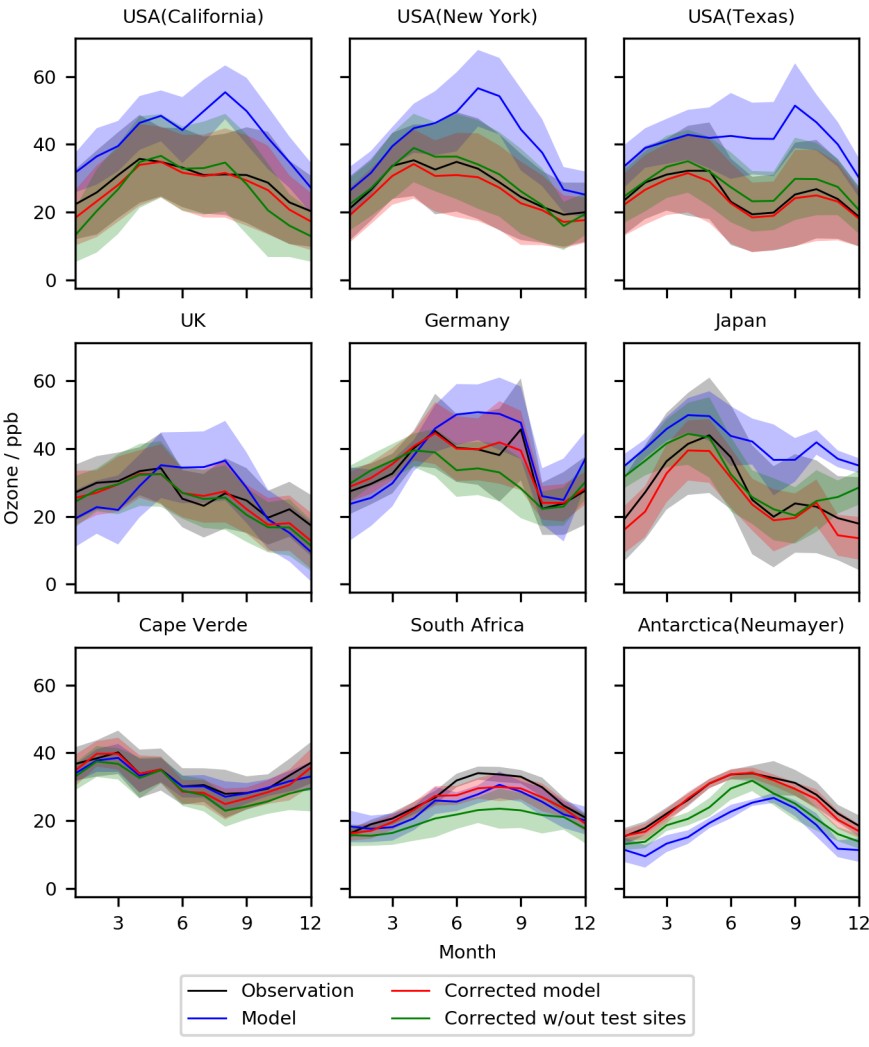

**Figure 10.** Seasonal cycle for O$_3$ at nine meta sites in 2016-2017. Shown are the observations, the base model and corrected model trained with all observations occurring at the nine sites removed. The median values are shown as the continuous line and the 25[th] to 75[th] percentiles as shaded areas.

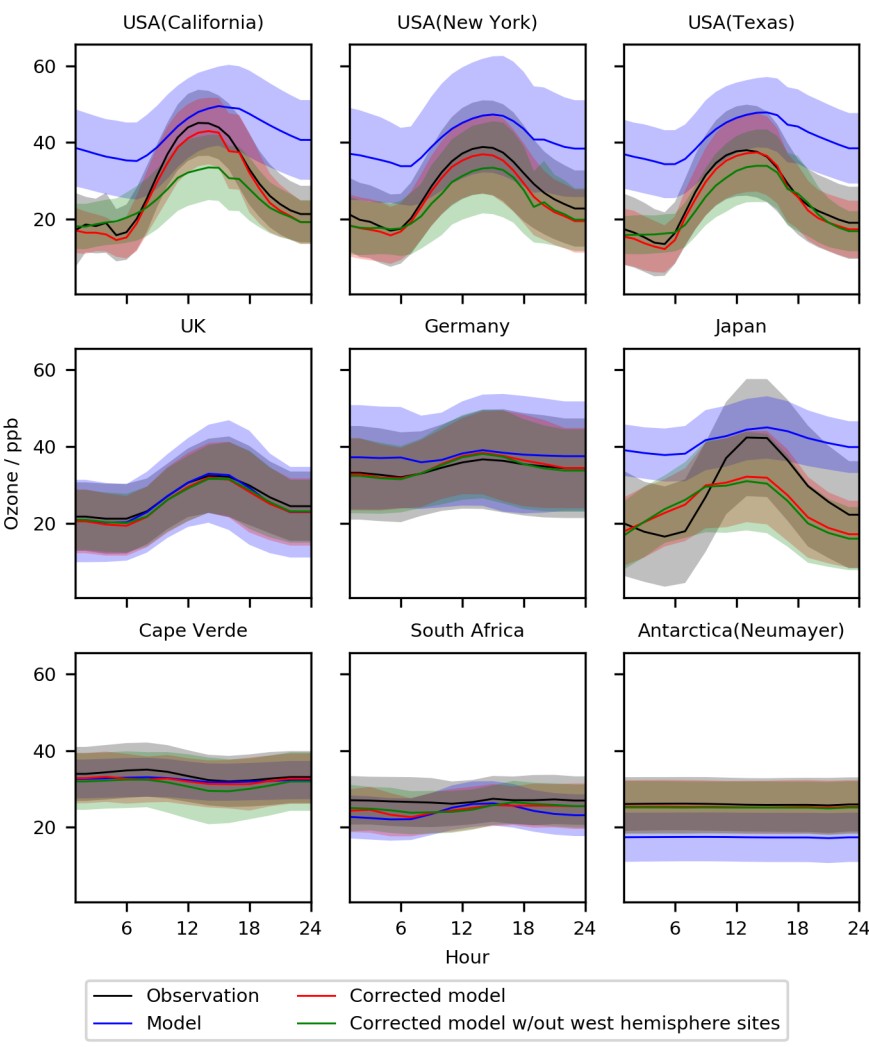

**Figure 11.** Diurnal cycle for O$_3$ at nine meta sites in 2016-2017. Shown are the observations, the base model, a corrected model trained using all of the observations and a corrected model trained with all western hemisphere (west of -20$^o$E) data removed. The median values are shown as the continuous line and the 25$^{th}$ to 75$^{th}$ percentiles as shaded areas.

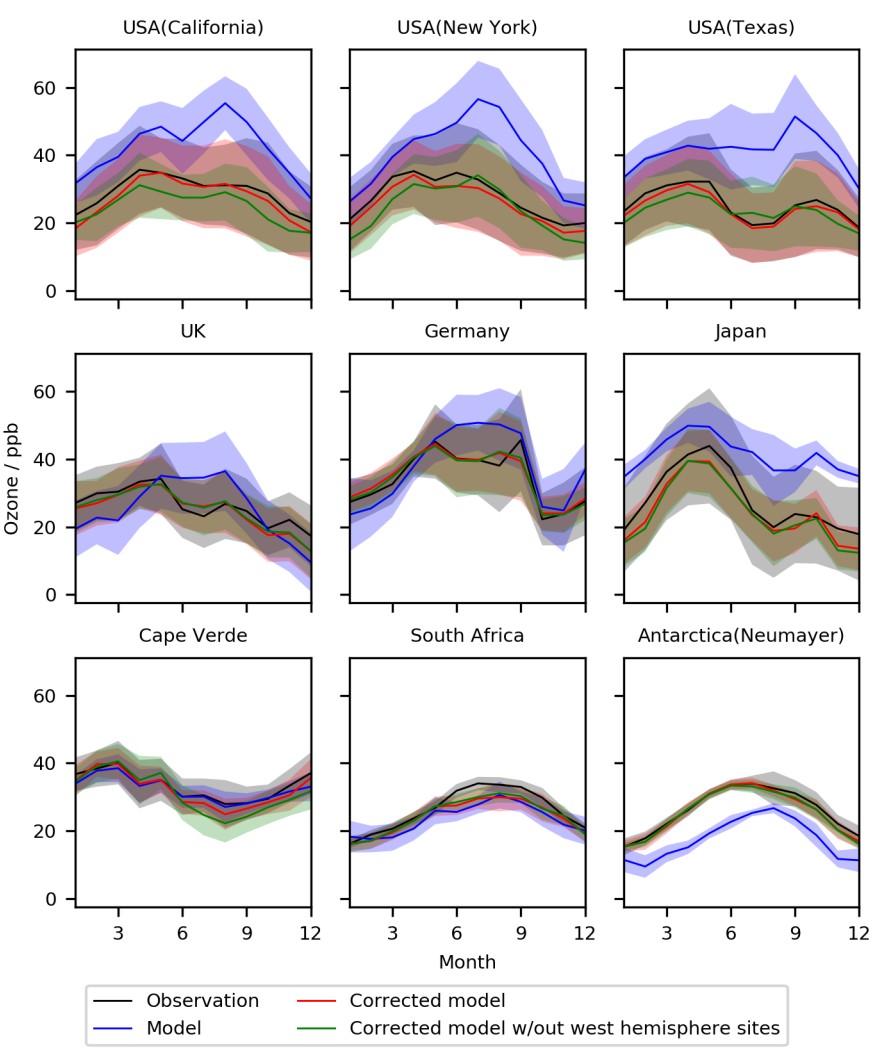

**Figure 12.** Seasonal cycle for $O_3$ at nine meta sites in 2016-2017. Shown are the observations, the base model, a corrected model trained using all of the observations and a corrected model trained with all western hemisphere (west of $-20^oE$) data removed. The median values are shown as the continuous line and the $25^{th}$ to $75^{th}$ percentiles as shaded areas.





**Figure 13.** Difference in the global mean annual surface O$_3$ prediction between a predictor trained with western hemisphere observation data (west of -20$^o$E) and a predictor trained without this data. Red dots show locations of ground sites in the surface to 900 hPa plot, and sonde locations in the other two plots.



**Table 1.** Chemical tracers and physical parameters used for training.

| Chemical tracers | | Physical parameters |
|---|---|---|
| NO | Hydrophilic black carbon | Pressure |
| $O_3$ | Hydrophobic organic carbon | Temperature |
| Peroxyacetylnitrate | Hydrophilic organic carbon | Absolute humidity |
| CO | 0.7 micron dust | Surface pressure |
| $\geq$C4 alkanes | 1.4 micron dust | Aerosol surface area |
| Isoprene | 2.4 micron dust | Horizontal wind speed |
| $HNO_3$ | 4.5 micron dust | Vertical wind speed |
| $H_2O_2$ | Isoprene epoxide | Surface albedo |
| Acetone | Accumulation mode sea salt aerosol | Cloud fraction |
| Methyl ethyl ketone | Coarse mode sea salt aerosol | Optical depth |
| Acetaldehyde | $Br_2$ | Solar zenith angle |
| $\geq$C4 aldehydes | Br | $Cos(day\,of\,year/360)2\pi$ |
| Methylvinylketone | BrO | $Sin(day\,of\,year/360)2\pi$ |
| Methacrolein | HOBr | |
| Peroxymethacryloyl nitrate | HBr | |
| Peroxypropionylnitrate | $BrNO_2$ | |
| $\geq$C4 alkylnitrates | $BrNO_3$ | |
| Propene | $CHBr_3$ | |
| Propane | $CH_2Br_2$ | |
| Formaldehyde | $CH_3Br$ | |
| Ethane | Methyl peroxy nitrate | |
| $N_2O_5$ | Beta isoprene nitrate | |
| $HNO_4$ | Delta isoprene nitrate | |
| Methylhydroperoxide | 5C acid from isoprene | |
| Dimethylsulfide | Propanone nitrate | |
| $SO_2$ | Hydroxyacetone | |
| $SO_4^{2-}$ | Glycoaldehyde | |
| $SO_4^{2-}$ on sea salt | $HNO_2$ | |
| Methanesulfonic acid | Nitrate from methyl ethyl ketone | |
| $NH_3$ | Nitrate from methacrolein | |
| $NH_4^+$ | Peroxide from isoprene | |
| Inorganic nitrates | Peroxyacetic acid | |
| Inorganic nitrates on sea salt | $NO_2$ | |
| Hydrophobic black carbon | $NO_3$ | |





**Table 2.** Statistics for diurnal profiles at the nine selected sites for the period 1/1/2016-31/12/2017, for the base model (BM), the model with the bias correction applied(BC), the corrector trained without the nine sites (NS) and the model trained without the western hemisphere data (NWH). Statistics are described in Sect. 5

| Site | Pearson's R | | | | RMSE / ppb | | | | NMB | | | |
|---|---|---|---|---|---|---|---|---|---|---|---|---|
| | BM | BC | NS | NWH | BM | BC | NS | NWH | BM | BC | NS | NWH |
| USA (California) | 0.852 | 0.997 | 0.986 | 0.983 | 14.74 | 1.98 | 3.59 | 6.57 | 0.46 | -0.06 | -0.11 | -0.15 |
| USA (New York) | 0.970 | 0.994 | 0.992 | 0.989 | 13.12 | 2.25 | 1.39 | 3.91 | 0.46 | -0.08 | 0.04 | -0.12 |
| USA (Texas) | 0.915 | 0.998 | 0.971 | 0.969 | 16.29 | 1.45 | 3.83 | 3.15 | 0.62 | -0.05 | 0.1 | -0.08 |
| UK | 0.993 | 0.998 | 0.998 | 0.998 | 1.02 | 1.39 | 2.29 | 1.13 | -0.02 | -0.05 | -0.08 | -0.04 |
| Germany | 0.791 | 0.991 | 0.982 | 0.973 | 3.25 | 0.92 | 2.88 | 0.81 | 0.09 | 0.01 | -0.07 | 0.0 |
| Japan | 0.98 | 0.764 | 0.949 | 0.648 | 14.9 | 6.94 | 5.46 | 8.03 | 0.48 | -0.12 | 0.12 | -0.14 |
| Cape Verde | 0.994 | 0.812 | 0.8 | 0.895 | 1.23 | 1.38 | 3.33 | 2.3 | -0.03 | -0.04 | -0.1 | -0.07 |
| South Africa (Cape Point) | 0.081 | 0.616 | -0.264 | 0.815 | 3.32 | 2.34 | 7.46 | 1.93 | -0.11 | -0.08 | -0.25 | -0.07 |
| Antarctica (Neumayer) | 0.883 | 0.872 | 0.532 | 0.73 | 8.57 | 0.67 | 4.75 | 0.85 | -0.33 | -0.03 | -0.18 | -0.03 |



**Table 3.** Statistics for seasonal profiles at the nine selected sites for the period 1/1/2016-31/12/2017, for the base model (BM), the model with the bias correction applied (BC), the corrector trained without the nine sites (NS) and the model trained without the western hemisphere data (NWH). Statistics are described in Sect. 4

| Site | Pearson's R | | | | RMSE / ppb | | | | NMB | | | |
|---|---|---|---|---|---|---|---|---|---|---|---|---|
| | BM | BC | NS | NWH | BM | BC | NS | NWH | BM | BC | NS | NWH |
| USA (California) | 0.833 | 0.987 | 0.952 | 0.948 | 14.02 | 2.19 | 5.2 | 4.54 | 0.45 | -0.06 | -0.11 | -0.15 |
| USA (New York) | 0.759 | 0.992 | 0.981 | 0.924 | 14.51 | 2.23 | 2.11 | 4.4 | 0.46 | -0.08 | 0.04 | -0.13 |
| USA (Texas) | 0.335 | 0.991 | 0.952 | 0.857 | 16.64 | 1.45 | 2.98 | 3.22 | 0.62 | -0.05 | 0.1 | -0.08 |
| UK | 0.519 | 0.935 | 0.939 | 0.939 | 7.27 | 2.51 | 3.11 | 2.27 | -0.03 | -0.05 | -0.08 | -0.04 |
| Germany | 0.848 | 0.956 | 0.663 | 0.963 | 6.55 | 2.42 | 6.37 | 2.13 | 0.09 | 0.01 | -0.07 | 0.0 |
| Japan | 0.939 | 0.972 | 0.812 | 0.968 | 14.0 | 3.92 | 6.34 | 4.59 | 0.48 | -0.12 | 0.13 | -0.14 |
| Cape Verde | 0.956 | 0.978 | 0.898 | 0.921 | 1.61 | 1.73 | 3.86 | 3.52 | -0.03 | -0.04 | -0.1 | -0.07 |
| South Africa (Cape Point) | 0.953 | 0.976 | 0.963 | 0.979 | 3.6 | 2.63 | 7.1 | 2.24 | -0.11 | -0.08 | -0.24 | -0.07 |
| Antarctica (Neumayer) | 0.939 | 0.993 | 0.968 | 0.993 | 8.86 | 1.04 | 5.02 | 1.14 | -0.33 | -0.03 | -0.18 | -0.03 |





**Table 4.** Statistical performance for the period 1/1/2016-31/12/2017 of the base model, model with a bias correction applied, and directly predicted $O_3$ concentration. Statistics are described in Sect. 4

| | Surface | | | | | | ATom | | | | |
|---|---|---|---|---|---|---|---|---|---|---|---|
| | RMSE | NMB | Pearson's R | Slope | Intercept | | RMSE | NMB | Pearson's R | Slope | Intercept |
| Base $O_3$ | 16.21 | 0.29 | 0.479 | 0.87 | 13.4 | | 12.11 | 0.08 | 0.761 | 1.15 | 2.73 |
| Corrected $O_3$ | 7.48 | -0.04 | 0.841 | 0.89 | 2.07 | | 10.50 | 0.06 | 0.792 | 1.00 | 2.28 |
| Predicted $O_3$ | 7.11 | 0.00 | 0.850 | 0.84 | 4.96 | | 10.92 | 0.11 | 0.797 | 1.01 | 3.69 |