# Peer review of "Improving the prediction of an atmospheric chemistry transport model using gradient boosted regression trees."

_Atmospheric Chemistry and Physics, 2019_

## Referee Comment (RC1) · Anonymous Referee #2 · 30 Oct 2019

The paper by Ivatt and Evans applies a machine learning technique for predicting biases in O3 simulations based on observations of O3 collected over previous time periods , and the statistical relationship between these and the model's chemical and physical state. The topic is interesting, timely, and suitable for this journal. The manuscript is generally clear and well written. My main concern is that is a little short / thin, particularly with regards to the relevance of this type of approach for air quality forecasting. The authors mention this numerous times, so it seems to be one of their prime motivating applications. However, from a lack of discussion of the background on this topic, to a lack of depth on exploration of the applicability of their methods to actually air quality forecasting needs, this aspect falls a bit short. The overall paper itself is on

the shorter side, so it seems with some revisions and substantial additions, the paper could become more applicable in this regard. That is is a "demonstration case" should not be grounds for incomplete context, analysis, or making claims beyond what has been actually shown.

Specific comments: The reference to Gaudel 2018 seems misplaced. The paper is nearly exclusively about trends in observed O3. In support of a statement regarding model biases, the authors are referred to the Young 2018 TOAR paper.

The introduction is too thin on the topic of O3 bias correction in models. There is a long, extensive history of O3 bias correction within the AQ literature. See for example Kang et al., 2010, https://doi.org/10.1016/j.atmosenv.2010.03.017, and half dozen or so papers cited in the introduction therein, and also additional research on the topic for more recent studies is warranted.

The authors bring up AQ forecasting frequently as an application. Concentrations and chemical environments relevant to forecasting seem to me much more highly variable at the scales of most forecasts (10's of km) compared to the analysis here (100's of km). How does that impact the authors conclusions regarding the applicability of their results? Would these techniques be expected to capture gradients in O3 biases between urban cores and surrounding areas? I don't see such issues presently discussed.

The observational dataset seems thin, particularly in Asia, given there is O3 data accessible there, through TOAR itself.

156: I don't really buy this explanation. The ML approach doesn't care if there is a true fundamental physical relationship, in reality. It only cares if there is a statistical relationship. The authors thus need to explain why a coastal cite degrades the statistical relationship. Further, I suspect the statistical relationship may be weak here owing to the importance of upwind sources in this region from China, which have a larger association with local O3 than the local model state.

[Figure]

Did the authors ever think about expanding the physical range of the model state that is included as input for the forecast in any one grid cell? This is done in the field of statistical prediction of PM, for example, since it is known that upwind conditions can drive local PM more than local conditions, especially when forecasting PM at high resolution. I would suspect the situation to be similar for O3. The present study may artificially benefit from the coarse model resolution not really resolving local O3 to begin with, but for future studies with high resolution models, this could become an important consideration.

How does the computational training time scale with the number of grid cells considered? This is an important consideration when considering the applicability of this approach to higher resolution simulations.

Fig 6 is great and left me wanting much more. Can the authors present this as well in terms of diurnal variability? Seasonal variability? Does the overall quantification of bias agree with biases noted in ensembles of air quality models in the Young 2018 TOAR paper (which seemed to have hemispheric N/S patterns, or is GEOS-Chem distinct?

The explanation for why O3 is an important predictor is a bit weak. I'm not sure I believe this is strictly an Antarctic / low-O3 result. But did the authors evaluate the spatial distribution of the importance of these predictors? That would certainly be interesting to see.

For the local model state, the authors didn't include land type or dry deposition velocity, which seem like would be important for correcting model biases associated with O3 loss, which is a known issue with these types of models.

Conclusions regarding the extent of data for training seem potentially biased by the way the authors have designed their performance metric. For AQ forecasting applications where the metric of performances is very short - term, it seems that training based on only the most recent conditions could be of more values, as indicated by the literature in that field.

[Figure]

It's a bit scary that removal of training data in areas like Cape Verde or South Africa make the predictions in these locations worse. Granted these locations are distinct from other areas, which performed fine or adequately when corresponding training data was removed. But my question is – how would we know, for a location with no training data – whether or not the bias corrections predicted here would help or degrade the simulation?

250: It seems like there could be regionally specific biases in meteorology that are not necessarily global in nature.

269 - 271: Not sure if I clearly understand the explanation being put forth here – could the authors expand?

277: The authors claim these methods offer a route to significant improvement in the fidelity of forecasts, but I'm not convinced the applicability to the priorities of air quality forecasting has really been demonstrated. We were not presented with any timeseries results. Nor was there evaluation of the extent to which this approach helped with prediction of exceedances or extreme values (or a reduction of false alarms). The results averaged over the entire coarse of the year tend to wash out the features that would be of most interest in a forecasting application.

Corrections: 102: tree, –> tree 132: the the 251: hemisphere( –> hemisphere ( 255: and so –> so 271: observed –> observations

---

## Referee Comment (RC2) · Anonymous Referee #1 · 3 Nov 2019

This paper describes the first application of a decision tree algorithm to predict the bias in an atmospheric chemistry transport model. It describes the approach taken, demonstrates its successful use for tropospheric ozone, and investigates its sensitivity to choice of training data and other options. The study is innovative in approach, and will certainly catalyse new research in application of machine learning techniques to atmospheric models. The paper is concise and well-structured, although there are some weaknesses in written style and presentation that need to be cleaned up. Overall the paper is appropriate for publication in ACP once the existing deficiencies have been addressed.

[Figure]

Specific Comments

Some drivers of model bias are likely to be non-local, particularly for longer-lived variables such as ozone. How might this be addressed within the framework of the current approach? Other biases will be due to limitations in how representative observations are of the scales resolved in the model (the comparison with coastal sites over Japan demonstrates this). How could these be addressed? It is notable that the observation data have already been filtered for urban and mountain sites. How sensitive are the results to the choice of which types of environment to exclude (distinct from the choice of location, which is covered well in section 7)?

l.59: "standard emissions configuration" It would help the reader to define standard (e.g., by citing a reference), or to drop this phrase if the subsequent list of inventories effectively covers it.

l.87: The phrase "most typical of" is unclear here, as flight measurements and sondes are different. The characteristics and sampling of the observations were similar?

l.98: "We have tended to favour" suggests prior work in this area, in which case it should be cited.

l.113: How is an "adequate level of complexity" defined for a particular problem?

l.115: A clause or sentence is required to define what "more interpretable" means in this context

l.157: The inability of the approach to correct the bias in some circumstances might also suggest that the parameters used for training are incomplete, and that some sources of bias are non-local.

l.191: The gain associated with a particular variable is not clearly defined here, and the feature importance (used on the y-axis in Fig 7) is undefined. Please add a sentence here to explain how these concepts are derived.

l.260: This final sentence presents an important but speculative conclusion, and would be much stronger if some evidence is provided from the study to back it up. The preceding analysis is vague about the reasons for the differences observed, so at least one concrete example of how model failings could be explained is needed.

l.264: How is the direct prediction implemented? Is model ozone used as one of the variables? If so, it appears unsurprising that "ozone + predicted (bias)" differs little from "predicted (ozone + bias)", and the main interest is therefore in what the differences tell us about the robustness of the decision algorithm used.

l.286: "Are all the variables needed?" It should be clear from the gain analysis shown in Fig 7 which variables are unimportant. If it isn't, why not, and if it is, what do you find?

l.294: The importance of particular variables is indicative not causative, and this would make it very difficult to extract information on the reasons for model biases. The night-time bias of the model is clear from a simple comparison with observations alone (Fig 2), and while identification of NO3 as important provides good evidence that the approach successfully recognises this, it is not clear that it is possible to reverse this process.

Numerical precision: one decimal place is sufficient for the biases, and two decimal places for the R values. Greater precision is not warranted in the abstract or body of the text.

The word "significantly" is used frequently in a colloquial sense rather than a statistical sense (e.g., l.174, l.177). For clarity, please use a different word or provide statistical metrics where appropriate.

Typos and minor corrections

l.14: shows -> show

l.30: "etc" better explained or removed

l.102: multi-variant -> multivariate

l.112: "this this"

l.115: underlies -> underlie

l.121: "5 kfold" -> "5 k-fold" or perhaps just "5-fold"

l.144: calculated the -> calculated with the

l.157: the reality -> reality

l.212: dot -> dots

---

## Short Comment (SC1) · 5 Nov 2019

Ivatt and Evans developed a gradient-boosted decision tree model (XGBoost) to correct the bias associated with GEOS-Chem-predicted O3 concentrations. They found that the bias-corrected model estimates ozone concentrations more accurately than the uncorrected model. The use of machine learning algorithms for air quality applications is a hot topic. I suggest that the following points need to be addressed to improve the quality of the manuscript.

1. This paper utilized the XGBoost model for bias correction of GEOS-Chem-predicted O3 concentrations. In the model training, the ground (EMEP, EPA, and GAW) and

ozone-sonde (WOURDC) observations were used. During the pre-processing of the training data set, the data comprising O3 concentrations above 100 ppbv were excluded. In general, a decrease in the maximum value of the target tends to increase the accuracy indicator of the machine learning model. However, the accuracy expressed in numerical values cannot always indicate the optimization level of the trained model. It is more logical to exclude the effects of stratospheric ozone using altitude information obtained from ozonesonde and ATom observations.

2. In the model development, the author presented only two important hyperparameters (depth and tree number) of the XGBoost model. One of the most important aspects of this study is the optimization of bias correction via logical development. Therefore, the results of sensitivity test conducted to determine important structural parameters (e.g., depth, number of trees, learning rate, tree boosting algorithm, and sub-sampling rate) should be provided.

3. In the model training, the K-fold algorithm was applied. The training and validation data set used observations from 2010 to 2015, which were divided into five blocks. The hyperparameters of the XGBoost model were determined by conducting the five independent model trainings. However, because the validation data sets were utilized K-1 times to optimize weight and bias matrix, the K-fold algorithm may be associated with the risk of training data leakage. Therefore, this issue should be addressed in the manuscript.

4. During the description of independent variables, the author only listed 81 variables as input features. In general, the accuracy of machine learning model is mainly attributed to the combination of input variables. In addition, imbalanced data set plays an important role in the performance of machine learning model. Therefore, it is necessary to use integrated analysis to identify the characteristics of the independent variables. Further, the close correlation between input features can distort the regression coefficients. In order to minimize the errors associated with munticorrelinearity, the author should provide a detailed analysis or rationale for the determination of input variables.

5. There is no detailed description of the observations. If the amount of the ground-based observations is much larger than that of the ozonesonde observations, it can affect the overall results of bias corrections. Therefore, the evaluation accuracy with ATom is clearly lower than that with the ground observations.

6. Data pre-processing has not been explained in detail. There are several ways involving data normalization. Several studies have suggested normalization methods to improve the performance of the predictive models. Optimization of data pre-processing methods should be addressed in the manuscript.

7. The grid resolution of GEOS-Chem was 4 İŁ x 5 İŁ. In order to prepare training and validation data sets, the observations were preprocessed to match with the grid of the GEOS-Chem. However, the limits on spatial representation of the observations are likely to cause sub-grid problems (i.e. the current grid size of GEOC-Chem is too large). The comparative results involving the Japan case clearly indicate the possibility of this problem. Therefore, it is necessary to develop a bias correction model with a higher resolution.

8. In this study, the importance of input variables was estimated based on gain. The importance of input features can be analyzed using various criteria (e.g. gain, weight and cover). Since these analyses can provide new scientific insights, a more comprehensive analysis of features based on various criteria is required.

---

## Author Comment (AC1) · 28 Feb 2020

**Review 1**

The reference to Gaudel 2018 seems misplaced. The paper is nearly exclusively about trends in observed O3. In support of a statement regarding model biases, the authors are referred to the Young 2018 TOAR paper.

We agree with the reviewer's suggested reference and have updated the paper accordingly.

The introduction is too thin on the topic of O3 bias correction in models. There is a long, extensive history of O3 bias correction within the AQ literature. See for example Kang et al., 2010, https://doi.org/10.1016/j.atmosenv.2010.03.017, and half dozen or so papers cited in the introduction therein, and also additional research on the topic for more recent studies is warranted.

We have added additional material here:

"Techniques used to reduce bias in air quality model include the use of ensembles (Wilczak et al., 2006) and data assimilation.Data assimilation techniques are used to incorporate observations into meteorological forecasts (Bauer et al., 2015) and some air quality models (Bocquet et al., 2015), techniques such a hybrid forecast (Kang et al., 2008; Silibello et al., 2015) or a Kalman filter (Delle Monache et al., 2006; Kang et al., 2010) have also been similarly applied."

The authors bring up AQ forecasting frequently as an application. Concentrations and chemical environments relevant to forecasting seem to me much more highly variable at the scales of most forecasts (10's of km) compared to the analysis here (100's of km). How does that impact the authors conclusions regarding the applicability of their results? Would these techniques be expected to capture gradients in O3 biases between urban cores and surrounding areas? I don't see such issues presently discussed.

We agree that the performance of this technique at the spatial resolution necessary for air quality forecasting is not specifically discussed and  have include additional text to emphasise this point:

"As forecast models are run at resolutions on the order of 1s - 10s kms, further work will need to be done to examine the technique's performance with the added variability associated with an increase in resolution. It is possible that some mitigation may be achieved with the inclusion of additional high resolution data, such as road usage or topological maps. Or with the use of variables that reflects the state beyond the grid-box (such as the concentration in adjacent boxes or the averaged of all boxes with in a varying range),to provide information on upwind conditions"

The observational dataset seems thin, particularly in Asia, given there is O3 data accessible there, through TOAR itself.

We agree that it would have been beneficial to have access to Asian data for our study. However, the data that is available from TOAR is often only the daily mean or other derived statistics due to data licensing issues. For many of the Asian sites, specific licenses are in place. As we show, much of the model failure is due to issues in the diurnal cycle; thus, we decided to use the smaller dataset of hourly observations rather than a potentially larger dataset of daily data.

156: I don't really buy this explanation. The ML approach doesn't care if there is a true fundamental physical relationship, in reality. It only cares if there is a statistical relationship. The authors thus need to explain why a coastal site degrades the statistical relationship. Further, I suspect the statistical relationship may be weak here owing to the importance of upwind sources in this region from China, which have a larger association with local O3 than the local model state.

We agree and have adapted the text to reflect this as a possible explanation for the issue over Japan:

"The Japanese data shows a differing pattern. Similar to the US sites, the base model over-estimates the O3 generating a much smaller diurnal cycle compared to the observation. Although the bias corrector improves the mean value, it does not completely correct the diurnal cycle. We attribute this to the coastal nature of Japan. The model grid-box containing the Japanese observations is mainly oceanic but the observations show a continental diurnal cycle (a marked increase in O3 During the day similar to those seen in the US). It is likely that the predicted bias is being distorted by biases at other ocean dominated grid-boxes, when in Japan's case, the O3 Concentration is likely influenced by long range transport from China"

Did the authors ever think about expanding the physical range of the model state that is included as input for the forecast in any one grid cell? This is done in the field of statistical prediction of PM, for example, since it is known that upwind conditions can drive local PM more than local conditions, especially when forecasting PM at high resolution. I would suspect the situation to be similar for O3. The present study may artificially benefit from the coarse model resolution not really resolving local O3 to begin with, but for future studies with high resolution models, this could become an important consideration.

We attempted to derive and explore a simple local bias correction methodology to outline the potential of this approach. Future work will look at exploring the required local and non-local information to improve the prediction. We have added a few sentences to the paper on this:

*"Or with the use of variables that reflects the state beyond the grid-box (such as the concentration in adjacent boxes or the average of all boxes within a varying range), to provide information on upwind conditions."*

How does the computational training time scale with the number of grid cells considered? This is an important consideration when considering the applicability of this approach to higher resolution simulations.

We have included additional text about the computational training-time scalability:

*"When altering the size of the training dataset we found the training time was approximately linear to the number of samples. For future high resolution runs we may consider the use of GPUs which have been found to substantially decrease training time (Huan et al., 2017)."*

Fig 6 is great and left me wanting much more. Can the authors present this as well in terms of diurnal variability? Seasonal variability?

We have now included a plot of the change in global diurnal and seasonal amplitude produced by the bias corrector and included a paragraph of discussion:

*"As we saw in the analysis of the nine individual sites, much of the improvement observed was due to the changes in the diurnal cycle. Figure 7 shows the global annual average change in diurnal cycle caused by the bias corrector. We see that there are only positive changes, increasing the amplitude of the diurnal cycle. This is likely due to the coarse model resolution not capturing the high concentration gradients required to achieve high rates of production or titration of O3. Conversely, Figure 8 shows that over polluted regions seasonal amplitude decreases. Which from the nine individual sites (Figure 3) appears to be a result of reductions in the predicted summer O3 Concentration."*

Does the overall quantification of bias agree with biases noted in ensembles of air quality models in the Young 2018 TOAR paper (which seemed to have hemispheric N/S patterns, or is GEOS-Chem distinct?

We have now added a free troposphere bias comparison to the paper:

*"In the free troposphere (900 to 400 hPa) we find the model is biased low in the southern extra-tropical and polar regions, and biased high in tropical regions. This Matches the pattern of the bias found at 500 hPa in the ensemble comparison performed in Young et al. (2018). However, that study found that the northern extra-tropical and polar region were biased low, whereas our results show a high bias, possibly due to a specific GEOS-Chem bias in these regions."*

The explanation for why O3 is an important predictor is a bit weak. I'm not sure I believe this is strictly an Antarctic / low-O3 result. But did the authors evaluate the spatial distribution of the importance of these predictors? That would certainly be interesting to see.

We agree and added that there might be added bias when the model goes into a period of unusually high or low ozone. The temporal and spatial variability in feature importance is an ongoing area of interest in future work we would look to retrain using only specific regions and examining the changes in feature importance. We have changed our explanation of the O3 feature importance in the paper:

"This feature appears to be being used to correct the concentration of O3 In Regions such as the US which are polluted and have a notably high bias at night. The next most important feature is the O3 concentration itself. This may be a result in biases arising during high or low O3 Periods. As well as reflecting biases in regions with very low O3 Concentrations such as around Antarctica."

For the local model state, the authors didn't include land type or dry deposition velocity, which seem like would be important for correcting model biases associated with O3 loss, which is a known issue with these types of models.

We agree that information about the land type could be an influential parameter. We did not include it in this paper but would take that forward to future work.

Conclusions regarding the extent of data for training seem potentially biased by the way the authors have designed their performance metric. For AQ forecasting applications where the metric of performances is very short - term, it seems that training based on only the most recent conditions could be of more values, as indicated by the literature in that field.

We have reduced emphasis on forecasting in the text. There is probably a need to separate "structural" error (i.e. rate constants and emissions) from "temporal" error (i.e. errors in the initial state). Future work will need to be done on this technique's potential to improve air quality forecasts.

It's a bit scary that removal of training data in areas like Cape Verde or South Africa make the predictions in these locations worse. Granted these locations are distinct from other areas, which performed fine or adequately when corresponding training data was removed. But my question is – how would we know, for a location with no training data – whether or not the bias corrections predicted here would help or degrade the simulation?

Given the current work, we are not in a position to predict whether the removal of information is likely to degrade the performance. Some consideration of this is necessary for future work. We have added some text to the paper to emphasise this point:

"While the algorithm is able to provide a prediction for any region, we can only have confidence in regions that we have test data."

250: It seems like there could be regionally specific biases in meteorology that are not necessarily global in nature.

As this is unknown we have changed our wording in the paper:

"This may be due to errors in the model's chemistry or meteorology, which could be global rather than local in nature."

269 - 271: Not sure if I clearly understand the explanation being put forth here – could the authors expand?

We have expanded the explanation in text:

"As XGBoost is unable to extrapolate outside the range of the observation data, direct prediction constrains to the observed O3 Concentration range. While this appears beneficial in areas we have observations, at sites where no observation training data isavailable, it is better to use the bias corrector approach as this only constrains the scale factor on the bias not the concentration itself."

277: The authors claim these methods offer a route to significant improvement in the fidelity of forecasts, but I'm not convinced the applicability to the priorities of air quality forecasting has really been demonstrated. We were not presented with any timeseries results. Nor was there evaluation of the extent to which this approach helped with prediction of exceedances or extreme values (or a reduction of false alarms). The results averaged over the entire coarse of the year tend to wash out the features that would be of most interest in a forecasting application.

We agree and reduced the emphasis on forecasting as discussed early in our response.

Corrections: 102: tree, –> tree 132: the the 251: hemisphere( –> hemisphere ( 255: and so –> so 271: observed –> observations

Corrected in the paper.

**Review 2**

Some drivers of model bias are likely to be non-local, particularly for longer-lived variables such as ozone. How might this be addressed within the framework of the current approach?

We agree and include some comments in the paper to reflect this.

"Or with the use of variables that reflects the state beyond the grid-box (such as the concentration in adjacent boxes or the average of all boxes within a varying range),to provide information on upwind conditions."

Other biases will be due to limitations in how representative observations are of the scales resolved in the model (the comparison with coastal sites over Japan demonstrates this). How could these be addressed?

Increasing the resolution is an ongoing area of investigation. We have added some sentences on how we would approach increasing the resolution:

"As forecast models are run at resolutions on the order of 1s - 10s kms, further work will need to be done to examine the technique's performance with the added variability associated with an increase in resolution. It is possible that some mitigation may be achieved with the inclusion of additional high resolution data, such as road usage or topological maps."

It is notable that the observation data have already been filtered for urban and mountain sites. How sensitive are the results to the choice of which types of environment to exclude (distinct from the choice of location, which is covered well in section 7)?

We agree with the reviewers suggestion and have added a couple of sentences in text:

"It would be possible to consider other data denial experiments based on site type (rural, industrial, residential, etc.) biome,altitude, etc. which could provide information about the utility of each observation. This would likely improve with running the base model at a higher resolution than was undertaken here."

l.59: "standard emissions configuration" It would help the reader to define standard (e.g., by citing a reference), or to drop this phrase if the subsequent list of inventories effectively covers it.

We have dropped the phrase in the paper.

l.87: The phrase "most typical of" is unclear here, as flight measurements and sondes are different. The characteristics and sampling of the observations were similar?

We have changed the phrasing to  "spatially similar".

l.98: "We have tended to favour" suggests prior work in this area, in which case it should be cited.

Changed wording to "Here we favour".

l.113: How is an "adequate level of complexity" defined for a particular problem?

We have added a sentence that describes point in more detail:

"This is then repeated until an adequate level of complexity is reached, where the model generalises the dataset without over-fitting."

l.115: A clause or sentence is required to define what "more interpretable" means in this context.

We have changed the sentence to be:

"he decision tree-based machine learning technique is more interpretable than neural net-based models (Kingsford and Salzberg, 2008), through the output of decision statistics"

l.157: The inability of the approach to correct the bias in some circumstances might also suggest that the parameters used for training are incomplete, and that some sources of bias are non-local.

As with the previous reviewers' comment on this, we have added a statement on the upwind contribution from China:

"It is likely that the predicted bias is being distorted by biases at other ocean dominated grid-boxes, when in Japan's case, the O3 Concentration is likely influenced by long range transport from China"

l.191: The gain associated with a particular variable is not clearly defined here, and the feature importance (used on the y-axis in Fig 7) is undefined. Please add a sentence here to explain how these concepts are derived.

The derivation of gain is rather complicated, and a sentence directing the reader to the paper containing the derivation has now been added:

"Derivation of gain metric for XGBoost can be found in Chen and Guestrin (2016)"

l.260: This final sentence presents an important but speculative conclusion, and would be much stronger if some evidence is provided from the study to back it up. The preceding analysis is vague about the reasons for the differences observed, so at least one concrete example of how model failings could be explained is needed.

This sentence has now been removed.

l.264: How is the direct prediction implemented? Is model ozone used as one of the variables? If so, it appears unsurprising that "ozone + predicted (bias)" differs little from "predicted (ozone + bias)", and the main interest is therefore in what the differences tell us about the robustness of the decision algorithm used.

We have added a more in-depth explanation on this text:

"As XGBoost is unable to extrapolate outside the range of the observation data, direct prediction constrains to the observed O3 Concentration range. While this appears beneficial in areas we have observations, at sites where no observation training data is available, it is better to use the bias corrector approach as this only constrains the scale factor on the bias, not the concentration itself."

I.286: "Are all the variables needed?" It should be clear from the gain analysis shown in Fig 7 which variables are unimportant. If it isn't, why not, and if it is, what do you find?

Merely removing a variable based on its low importance is difficult. As many variables are correlated, and the removal of one can result in another variable being used to identify the same relationship. i.e. removing a compound present at night results in another night-time species increasing in importance resulting in only a small drop in performance. Removing a less important variable may have no correlated variable and result in a more significant reduction in performance as information is lost.

I.294: The importance of particular variables is indicative not causative, and this would make it very difficult to extract information on the reasons for model biases. The night-time bias of the model is clear from a simple comparison with observations alone (Fig 2), and while identification of NO3 as important provides good evidence that the approach successfully recognises this, it is not clear that it is possible to reverse this process.

We have changed the wording in the paper:

"While much of the information provided by the predictor is indicative rather than causative, coupling feature importance and data denial with domain knowledge may provide a powerful diagnostic technique for identifying the source of bias."

Numerical precision: one decimal place is sufficient for the biases, and two decimal places for the R values. Greater precision is not warranted in the abstract or body of the text.

We have reduced the precision of RMSE and R values as recommended.

The word "significantly" is used frequently in a colloquial sense rather than a statistical sense (e.g., I.174, I.177). For clarity, please use a different word or provide statistical metrics where appropriate.

We have replaced the word significantly throughout the paper.

Typos and minor corrections I.14: shows -> show I.30: "etc" better explained or removed I.102: multi-variant -> multivariate I.112: "this this" I.115: underlies -> underlie I.121: "5 kfold" -> "5 k-fold" or perhaps just "5-fold" I.144: calculated the -> calculated with the I.157: the reality -> reality I.212: dot -> dots

Corrected in text.

**Comments**

1. This paper utilised the XGBoost model for bias correction of GEOS-Chem-predicted O3 concentrations. In the model training, the ground (EMEP, EPA, and GAW) and ozone-sonde (WOURDC) observations were used. During the pre-processing of the training data set, the data comprising O3 concentrations above 100 ppbv were excluded. In general, a decrease in the maximum value of the target tends to increase the accuracy indicator of the machine learning model. However, the accuracy expressed in numerical values cannot always indicate the optimisation level of the trained model. It is more logical to exclude the effects of stratospheric ozone using altitude information obtained from ozonesonde and ATom observations.

The use of 100 ppb as a definition of the stratosphere is well characterized in the literature. We provide a reference to that approach. The tropopause does not occur at a single value of altitude, and it varies systematically with season and latitude, and also in response to meteorological forcing. A simple height based assessment would not work. Hence we have used a chemical tracer approach.

2. In the model development, the author presented only two important hyperparameters (depth and tree number) of the XGBoost model. One of the most important aspects of this study is the optimisation of bias correction via logical development. Therefore, the results of sensitivity test conducted to determine important structural parameters (e.g., depth, number of trees, learning rate, tree boosting algorithm, and sub-sampling rate) should be provided.

In this paper, we were aiming to show that it was possible to derive a local bias correction methodology based on modern machine learning techniques. Future development and evaluation will explore the sensitivity to hyperparameters.

3. In the model training, the K-fold algorithm was applied. The training and validation data set used observations from 2010 to 2015, which were divided into five blocks. The hyperparameters of the XGBoost model were determined by conducting the five independent model trainings. However, because the validation data sets were utilised K-1 times to optimise weight and bias matrix, the K-fold algorithm may be associated with the risk of training data leakage. Therefore, this issue should be addressed in the manuscript.

In this paper, we were aiming to show that it was possible to derive a local bias correction methodology based on modern machine learning techniques. Future development and evaluation will explore the sensitivity to training methodology.

4. During the description of independent variables, the author only listed 81 variables as input features. In general, the accuracy of machine learning model is mainly attributed to the combination of input variables. In addition, imbalanced data set plays an important role in the performance of machine learning model. Therefore, it is necessary to use integrated analysis to

identify the characteristics of the independent variables. Further, the close correlation between input features can distort the regression coefficients. In order to minimise the errors associated with munticorrelinearity, the author should provide a detailed analysis or rationale for the determination of input variables.

Most of the 81 variables are those transported chemical species in the model. The others reflect key meteorological variables. Future development and evaluation will explore the choice of input parameters.  We have updated the text to explain this.

5. There is no detailed description of the observations. If the amount of the groundbased observations is much larger than that of the ozonesonde observations, it can affect the overall results of bias corrections. Therefore, the evaluation accuracy with ATom is clearly lower than that with the ground observations.

We come to the same conclusions and have stated this in the paper.

6. Data pre-processing has not been explained in detail. There are several ways involving data normalisation. Several studies have suggested normalisation methods to improve the performance of the predictive models. Optimisation of data pre-processing methods should be addressed in the manuscript.

We have described the processing of the data. With the XGBoost algorithm used here (unlike say neural nets) no normalization is needed.

7. The grid resolution of GEOS-Chem was 4 x 5. In order to prepare training and validation data sets, the observations were preprocessed to match with the grid of the GEOS-Chem. However, the limits on spatial representation of the observations are likely to cause sub-grid problems (i.e. the current grid size of GEOC-Chem is too large). The comparative results involving the Japan case clearly indicate the possibility of this problem. Therefore, it is necessary to develop a bias correction model with a higher resolution.

In this paper we were aiming to show that it was possible to derive a local bias correction methodology based on modern machine learning techniques. Future work will explore the influence of spatial resolution.

8. In this study, the importance of input variables was estimated based on gain. The importance of input features can be analysed using various criteria (e.g. gain, weight and cover). Since these analyses can provide new scientific insights, a more comprehensive analysis of features based on various criteria is required.

In this paper we were aiming to show that it was possible to derive a local bias correction methodology based on modern machine learning techniques. Future development and evaluation in future publications will explore the potential for diagnostic metrics to enhance our understanding of the processes.

---

## Author Response (AR2)

Dear Handling Editor,

I have altered the manuscript based on the grammatical recommendations of the reviewer, with the differences attached below.

I found the suggestion of more significant revisions but with the manuscript only open for short minor corrections a little confusing.  I am about to submit my PhD thesis and unfortunately do not have time to undertake a major update of the work.

With the corrections that have now been made I would ideally like the paper to be submitted for publication.

Thank you,

Peter

[revised manuscript text omitted]